# Learning to Be an Orangutan—Implications of Life History for Orangutan Rehabilitation

**DOI:** 10.3390/ani11030767

**Published:** 2021-03-10

**Authors:** Signe Preuschoft, Ishak Yassir, Asti Iryanti Putri, Nur Aoliya, Erma Yuliani, Siti Nur Badriyah, Paloma Corbi, Yoyok Sugianto, Bina Swastas Sitepu, Elfriede Kalcher-Sommersguter

**Affiliations:** 1Ape Protection Unit, Four Paws, 22767 Hamburg, Germany; paloma.corbi@four-paws.org; 2Yayasan Jejak Pulang, Samboja, East Kalimantan 75276, Indonesia; asti2007044029@webmail.uad.ac.id (A.I.P.); nuraoliya@apps.ipb.ac.id (N.A.); ermayulianisaputri@gmail.com (E.Y.); sitinurb94@gmail.com (S.N.B.); 3Balitek KSDA, Ministry of Environment and Forestry, Samboja, East Kalimantan 75276, Indonesia; Ishak.Yassir@gmail.com (I.Y.); binassitepu@gmail.com (B.S.S.); mukhlisi_arkan@gmail.com (M.); 4Department of Psychology, Ahmad Dahlan University, Yogjakarta 55166, Indonesia; 5Department of Biology, IPB University, Bogor 16680, Indonesia; 6BKSDA Kalimantan Timur, Ministry of Environment and Forestry, Samarinda, East Kalimantan 75243, Indonesia; yoyokbksda@gmail.com; 7Institute of Biology, University of Graz, 8010 Graz, Austria

**Keywords:** rehabilitation, re-introduction, ontogeny, orphans, orangutan, life history, allo-mothering, peer-rearing

## Abstract

**Simple Summary:**

Like humans, great apes have extended childhoods during which they depend on maternal pedagogy. To help rescued orphans recover from maternal loss our rehabilitation programme is modelled on the natural infant development of orangutans. Orphaned apes cannot be released back into freedom before they have learned the skills necessary to survive alone. To prevent suffering after release we documented the development of survival skills during the rehabilitation process. Seven orangutan orphans aged 1.5–9 years were observed over 18 months in their forest school, immersed in a natural forest environment with human surrogate mothers and other orphans. Social interactions deviated from wild mother-reared immatures: Infant orphans spent more time playing with peers, rested less, and were far from their human surrogate mothers earlier and often than wild immatures are from their biological mothers. Around weaning age, 4- to 7-year-old orphans took up a typical orangutan life-style: they interacted less with human surrogate mothers and peers, stayed high in the trees and slept in nests in the forest. Their time budgets resembled those of wild adults. We conclude that it is not only ethical but also possible to assess survival competences of rehabilitant orphans before release and choose release candidates accordingly.

**Abstract:**

Orangutans depend on social learning for the acquisition of survival skills. The development of skills is not usually assessed in rescued orphans’ pre-release. We collected data of seven orphans over an 18-months-period to monitor the progress of ontogenetic changes. The orphans, 1.5–9 years old, were immersed in a natural forest environment with human surrogate mothers and other orphans. Social interactions deviated significantly from those of wild mother-reared immatures. Infants spent more time playing socially with peers, at the expense of resting and solitary play. Infants were also more often and at an earlier age distant from their human surrogate mothers than wild immatures are from their biological mothers. We found important changes towards an orangutan-typical lifestyle in 4- to 7-year-old orphans, corresponding to the weaning age in maternally reared immatures. The older orphans spent less time interacting with human surrogate mothers or peers, started to use the canopy more than lower forest strata and began to sleep in nests in the forest. Their time budgets resembled those of wild adults. In conclusion, juvenile orphans can develop capacities that qualify them as candidates for release back into natural habitat when protected from humanising influences and immersed in a species-typical environment.

## 1. Introduction

Worldwide, unsustainable population and economic growth is resulting in destruction of natural balances and extinction of species. Large-bodied long-living species such as non-human apes are victims of competition with their human cousins for land. Among the great apes, orangutans (*Pongo* spp.) have been the first to slide over the brink of the extinction threshold when mortality rates can no longer be offset by reproduction [1]. This situation poses novel challenges to conservation where preventive protection of still intact biotopes is no longer enough to guarantee the continued survival of species and their web of interconnected coexistence. Therefore, conservation by restorative interventions is becoming increasingly important.

Raising ape orphans to become self-sufficient is cost-intensive and complicated due to their slow ontogenetic development and high propensity for social learning. Orangutans have very slow life histories: after a gestation period of eight months, a single altricial offspring is born. This offspring is only weaned after 5–8 years. Female age at first birth is high (14.8 years), as is female survival until first reproduction (94 percent), inter-birth intervals are long (7.6 years), infants survive at extremely high rates (91 percent) [2]. Slow life histories (or K-selection) tend to be associated with longevity and behavioural flexibility, i.e., less dependence on instinct and increased importance of learning and cognition. Orangutan mothers continue to invest into their offspring postnatally for years: babies are in body contact almost all the time until they reach two years of age, infants share their mother′s nests until weaning, and full locomotory independence is only reached at 5–6 years. In this species where feeding competition is so intense that it leads to a solitary life-style, infants stay close to their mothers and eat what mothers eat thus learning which foods to select, where and when to find it, and how to extract edible parts from embedded foods [3]. 

Great ape rescue and rehabilitation with the purpose of later release back into natural habitat has been attempted since the unsustainable exploitation of tropical resources began to accelerate after the Second World War [4]. In Indonesia, the first such programmes coincided with the start of field research into the behaviour of wild-living orangutans [5,6,7]. Meanwhile, the study of captive apes led to immense progress in psychological and ethological understanding of the nature of apes and their profound similarities with the human ape [8,9,10,11]. However, these insights have not yet been fully utilised to inform great ape rehabilitation for later release. For instance, the programmes have been releasing orphans back into the wild years before their natural age of weaning from maternal lactation at 5–7 years. A frequently cited review found that several years after their release 20–80 percent of orangutans had vanished [12]. While practitioners still debate whether vanished orangutans can count as re-introduction successes, it is obvious that one profound weakness of releases conducted before the turn of the millennium is poor record keeping. For instance, it is not possible to get a clear number of the orangutans actually released. This is not inconsequential, as the number could be in the range of 1500 orangutans who have been released between 1971 and 2020 (1227 released rehabilitants between 1971 and 2013 according to [13]). A pessimistic estimate would therefore lead to an estimated number of 1200 orangutans who disappeared, i.e., who might be dead, after release.

A recent evaluation suggests that such a lack of transparency persists [14]. Of 605 rehabilitated orangutans released between 2007 and 2017, no detailed information was available to the reviewers for 29 percent. On the remaining 71 percent of released orangutans, no systematic information was available either. This suggests that—practical problems aside—practitioners are unclear about the parameters that need to be recorded to permit evaluation and quality control [15]. In view of the currently available knowledge about great ape ontogenetic development, learning and social psychology this ignorance is difficult to understand (or excuse). 

Among the 431 orangutans released between 2007 and 2017, for which Sherman et al. [14] were provided with more detailed information, were 16 percent (69 individuals) who needed to be re-rescued after their release. Among the top reasons for re-rescue were injury of the orangutans, human-orangutan conflict (e.g., crop eating by the orangutans, harassment/attack/capture of orangutan by humans), and inability to adjust to natural life (incl. starving/being underweight). Multiple re-capture and re-release was conducted for 10 individuals (2.3 percent). It is obvious that having information pre-release about the survival competences of rehabilitant orangutans would be beneficial for the wellbeing of the orangutans concerned and also represent a more efficient use of funding. Aware of these shortcomings, the Indonesian ministry of forestry initiated in 2017 a new programme for orangutan rehabilitation and release that is expected to remedy some of these weaknesses. This programme is conducted in partnership between the Indonesian Ministry of Forestry and Environment, represented by Balitek KSDA Samboja, BKSDA Samarinda, and Yayasan Jejak Pulang, an Indonesian foundation that cooperates with the animal welfare organisation Four Paws. 

In this article, we present the first results from data collected in the course of rehabilitation, i.e., pre-release. By collecting systematic observational information, we intend a) to gain a deeper understanding of parameters and rehabilitation measures that enable an orphaned orangutan to successfully adapt to life in freedom in the natural habitat, and thus b) to assess what is needed to make rescue, rehabilitation, and re-introduction of apes into a successful and effective conservation tool.

To ensure comparability of our data, behaviour recording followed standardised data collection procedures and categories established by field workers for post-release monitoring [15,16]. To complement assessment of skill acquisition and ecological performance we added behaviour elements indicative of immatures′ socio-emotional development. Another important innovation was the use of tablets for data recording in the field. This allowed to shortcut the tedious and time-consuming manual entry of paper-and-pencil data into a computer. Since 2018, we have trained 13 caregivers to record data in this way, and their inter-observer reliability is tested every six months by either co-observing or use of novel video material.

Data are analysed monthly and summarised quarterly to enable assessments of trends over time. The results of these analyses are discussed with caregivers and among a team of ethologists and veterinarians in order to agree on rehabilitation or therapeutic interventions on a case-by-case basis within a time frame when management changes can still benefit the orphan concerned, i.e., long before a decision on her or his suitability for release.

Here, we present data collected over a period of 18 months (January 2019—June 2020), to determine behaviour changes over time. We assume such changes may be caused by two factors:There could be progressive changes that are likely to be caused by maturation and learning processes of the orphaned orangutans, as they get older.A second possible cause of changes over time is the seasonal variation (cyclical) within the habitat, such as fluctuations in food availability.

We also expect interactions between these factors in so far, e.g., as very young orangutan orphans may not be able to adjust their behaviour to phenological variation because they do not yet have the knowledge, skill, or physical power to utilise all available food sources. The influence of phenological variation is not analysed in this paper.

## 2. Subjects, Management and Methods

The “forest school” is the place where rehabilitant orangutans are prepared for re-introduction, after having first passed incoming quarantine and health exams. The forest school (Figure 1) consists of ca. 100 ha of low-land forest and is part of a larger area of legally protected forest (Tahura), which also includes the study forest of Balitek Samboja (KHDTK). The area was affected in various degrees by illegal logging and forest fires in 1998 and has regrown relatively undisturbed since that time. 

In order to avoid “humanising” the rehabilitant orphans we are using several principles in combination.
(a)To align the rehabilitation measures with the natural ontogeny of mother-reared wild orangutans. This implies to avoid abrupt changes in living conditions or animal management.(b)To “orangutanise” the human caregivers instead of humanising the orangutan orphans. This implies to immerse the orphans in an orangutan-typical environment as soon as they can leave incoming quarantine, and to minimise exposure to human objects, lifestyle, communication, and culture (see segregation of areas in Figure 1).(c)To maintain a strict non-contact policy which prevents any contact between orangutan orphans and any human person not directly involved with orangutan care.

### 2.1. Subjects

Yayasan Jejak Pulang (YJP) currently cares for and rehabilitates 10 orangutans. Here, we present data on seven orangutans, who had passed quarantine (Table 1). The seven rehabilitants were managed in two groups with differing daily routines. Four infants (Forest School level 1, FS1) were managed in the Eastern area of forest school, with two shifts of caregivers, and a ratio of ca. 1.3 orangutan: 1 caregiver, varying between 3 and 4 caregivers per day for the 4 infants. The infants spent around 11 h per day in the forest and slept indoors onsite (Figure 2).

The three older orangutans used a larger area, with night cages at Sungai Sakakanan, and a feeding platform well inside the forest school (Forest School level 2, FS2). To these older orphans we refer as “juveniles” although, strictly speaking, Cantik and Eska are still in pre-weaning age [17]. From June 2019, Cantik, the youngest rehabilitant in FS2, started to sleep in the forest. Amalia, who joined Eska and Cantik in July 2019, began sleeping in the forest right away (Forest School level 3, FS3, see Figure 2). These “juveniles” were managed in three shifts starting at 5:30 in the morning. At noon, the second day shift replaces the morning team, handing over to the night shift at 19:00. During the daylight hours in each shift there is ratio of 1 orangutan: 1 caregiver, because the older orangutans often range independent from each other.

### 2.2. Data Collection

Individual orangutans were observed using instantaneous scan sampling with intervals of 2 min. The data collection method and categories of the activity budget (feeding, resting, travelling and social) followed standardised protocols for field research and post-release monitoring (cf. [15,18]). Each juvenile orangutan was observed for a total of three full observation days per month. Each infant was observed for a total of two full days per month. Observation periods lasted 1 h and were distributed evenly over the 12 h daylight between 6:00 and 18:00 (juveniles) or 7:00 and 17:00 (infants).

We trained a total of 13 caregivers over a period of 20 months to collect the behavioural data with an inter-observer reliability (IOR) of >80%. To test for IOR ten videos are shown to the caregivers where they have to assign the height of the orangutan, the behavioural category and respective subcategories correctly. Inter-observer reliability is re-assessed roughly every 6 months. Data from caregivers who do not reach the 80% criterion are excluded. Data are entered into electronic devices, using ZooMonitor [19]. After entering the observations, the data are saved in the device, and up-loaded to the internet once every two weeks, so there is minimal loss of data due to lack of discipline or adverse weather. Each month, the data are carefully edited by an Ethology Assistant who will approach caregivers for clarification and corrections. The Ethology Assistant will then analyse the data according to a routine procedure and provide simple graphs for the monthly report. These routine analyses include the activity budget, use of forest stratum, type of food and part of plant eaten as well as proximity to caregivers and conspecifics. The monthly report needs to be discussed between the management and the caregivers to streamline the feedback process and agree on orangutan management decisions. 

The data collection focuses on ecological competences (e.g., feeding skills, nest building, rain protection), time budgets and social parameters indicative of orphan′s development towards independence. Human caregivers were included as interaction partners because they function as surrogate mothers to the orangutan orphans (see Appendix A for ethogram).

Here we present data pooled over the 18 months period (January 2019–June 2020), and also grouped data into six quarters of three months, starting with January–March 2019 (Q1) and ending with April–June 2020 (Q6, Table 2). Data are analysed per individual, to be able to detect developmental changes within each subject. The small sample sizes (N = 3 juveniles and N = 4 infants) limited the statistical analyses to Pearson Chi-Square tests to test for inter-individual differences between juveniles and infants as well as for intra-individual differences across the six quarters.

## 3. Results

### 3.1. Activity Budgets

In field studies of wild orangutans, usually four categories of activity are distinguished: feeding, resting, travelling and “other” which includes social behaviour, tool use and other relatively rare activities [20]. Since our orphans have more opportunities for social interactions, because they are managed either together in groups (infants) or live at much higher densities and encounter rates than wild age-mates (juveniles), we have separated social behaviour from the “other” category in our analyses (Figure 3).

All seven rehabilitating orphans spent most of their time feeding (range: 34–52%). Travelling was the second most frequent activity (range: 21–25%), and resting ranked third, taking up 6–13% of their activity budget. Only Amalia rested more than she travelled (rest: 22%, travel: 21%). As expected, the rehabilitant orphans spend much time on social activities (8–23%), which may rival resting time. The exception is again the oldest rehabilitant Amalia who spent only about 1% of her time on social activities. Thus, eight-year-old Amalia’s activity budget deviated from the other rehabilitants and resembled that of wild adult orangutans with a predominance of feeding, followed by resting and travelling, and only very small portions of social interactions and other behaviours.

An inter-individual comparison revealed that the activity budgets differed significantly (Pearson Chi Square Test: χ^2^ = 6243.3, df = 24, *p* < 0.001). The percent of scans spent on feeding was highest in the juvenile females Amalia and Cantik (Figure 3, over 50%) and significantly exceeded expected values. By contrast, all infants were observed feeding between 34 and 42% of their time. Percent of scans spent on feeding of juvenile male Eska and three of the four infants were significantly lower than expected (see Table 3).

Inter-individual differences were found with respect to resting: Amalia and Kartini rested more than expected, Cantik and the infant males rested significantly less than expected (Table 3).

Inter-individual differences were also found with respect to travelling: All three females travelled significantly less than expected, the males, except Gerhana, significantly more than expected (Table 3).

Social interactions suggest an age or maturation effect. In juveniles, social interactions made up significantly less of the activity budgets, but exceeded expected values in the infants, except Kartini, the oldest infant (Table 3). 

This age or maturation effect is not only indicated by the comparisons between individuals but also traceably on the intra-individual level when we display relative proportions of resting and social interactions over time (Figure 4a,b). Starting in Q4, until Q6 juveniles Amalia, Eska and Cantik decreased their social interactions and increased their resting time (see Figure 4a). The infants, by contrast, continued to show high rates of social activities throughout the observation period (Figure 4b).

### 3.2. Interaction Partners: Human vs. Conspecific

The higher proportion of social interaction in the activity budgets of infants (13–23% compared to 1–10% for the juveniles; Figure 5 might reflect their higher need for attachment and protection from a substitute mother. We analysed how much this social interaction time was allocated to human caregivers and conspecific peers, respectively.

Over the course of the study period, the juveniles decreased their interaction frequencies with other orangutans (Figure 6a), while interactions with human caregivers were infrequent throughout (Figure 7a). From age 5 to 7 years Eska shows a consistent trend to spend less and less time on social interactions. Likewise, Cantik starts to decrease her social interaction frequencies from Q4, when she was ca. 5 years old, shortly after beginning to sleep in night nests in the forest (Q2/3). 8-year-old Amalia started out at a lower frequency of social interactions, but by Q6, both Eska and Amalia have almost no social interactions any more. These developmental trends are significant (see Appendix A). Cantik, however, exhibited no clear pattern regarding her interactions with human caregivers, whereas her interactions with peers did decrease gradually (Figure 6a, Appendix A; Amalia: χ^2^ = 76.7, df = 4, *p* < 0.001; Eska: χ^2^ = 648.7, df = 10, *p* < 0.001; Cantik: χ^2^ = 674.8, df = 10, *p* < 0.001). Thus, the interaction patterns of the three juveniles begin to resemble those of wild adolescent and adult orangutans. For the infants no such trend was apparent until Q6, but interaction rates of the oldest infant Kartini with humans declined (Figure 7b, yellow), similarly to those of the juveniles, again suggesting a maturation effect.

Infants interacted just a little less with their human caregivers (5–11%) than with each other (8–15%; Figure 5), whereas juveniles spent more time interacting with each other (1–9%) than with human caregivers (0.5–2%). Displaying the individual proportions spent on social interaction over time confirms the impression of a maturation effect:

### 3.3. Proximities to Different Classes of Social Partners

Overall, infants spent more time within arm′s reach of a caregiver, and were rarely in more than 10 m distance of an allo-mother (Figure 8). Juveniles, by contrast, spent most of their time more than 5 m away from caregivers, and were virtually never within arm’s reach.

Regarding close proximity, infant orphans preferred female over male surrogate mothers. They were observed significantly more often within 1 m of a female than a male caregiver (Pearson Chi Square Test: Kartini: 6.7 vs. 3.2%, χ^2^ = 145.0, df = 3, *p* < 0.001; Tegar: 12.1 vs. 6.4%, χ^2^ = 157.9, df = 3, *p* < 0.001; Gonda: 17.0 vs. 2.9%, χ^2^ = 772.0, df = 3, *p* < 0.001; Gerhana: 8.4 vs. 6.7%, χ^2^ = 49.8, df = 3, *p* < 0.001). Like the infants, 5 year-old Cantik was significantly more often within 1 m of female than male caregivers (4.4 vs. 1.6%, χ^2^ = 447.3, df = 3, *p* < 0.001), but she also was farther than 10 m from any caregiver over 40% of the time.

Similarly, Eska was more than 10 m distant from female or male caregivers over 27% of his time. Most extreme was Amalia, the oldest, who was only about 1% of her time within 1 m of a caregiver (female 0.6% vs. male 1.1%) and about 10% of her time within 5 m distance of a caregiver (female 9.1% vs. male 9.8%), but spent most of her time further than 10 m away from caregivers. In this she discriminated against female caregivers and was more tolerant of men by spending 77.0% of scans out of 10 m to female caregivers vs. 65.6% to male caregivers (χ^2^ = 124.9, df = 3, *p* < 0.001). Eska was observed significantly less often within 1 m of female than of male caregivers (0.4 vs. 2.6%, χ^2^ = 175.6, df = 3, *p* < 0.001), because female caregivers avoided being close to him so as not to get involved in play-wrestling.

Infant orphans were close to one another more often than were juveniles. They spent more than 80% of their time within 10 m from one another (Table 4), similar to their proximity with surrogate mothers. Larger distances of over 10 m from other infants occurred in only 12–19% of scans. Juveniles, by contrast, spent 45–76% of scans more than 10 m away from their conspecifics (Table 5).

### 3.4. Use of Forest Stratum

All orphans spent time on the ground, but all of them also climbed around in over 10 m height, even the youngest. The forest school has many tall trees with heights well above 10 m, but for comparability with areas with smaller trees we used 10 m and more as the highest category for coding. As expected, juveniles spent less time on the ground (mean: 24%, range: 7–39%) and more time high in the canopy (>10 m: mean: 37%, range: 20–62%) than did infants (on ground mean: 40%, range: 37–43%), >10 m mean: 15%, range: 14–16%). This difference in use of the highest forest stratum is significant (mean percent of scans in >10 m per quarter (Q1–Q6), Mann-Whitney U-test: U = 6.0, one-tailed *p* = 0.035). Again, this age effect comes into sharper focus when we look at the intra-individual development. From Q4, juveniles consistently increased their time in the highest stratum category (above 10 m, Figure 9a–c). A comparison of the first (Q1–Q3) with the second half (Q4–Q6) of the study period was precluded by small sample size. By contrast, no such trends were visible in the infants. All infants continued to spend more time below 5 m height than above (means across all infants: Figure 9d).

Amalia was high up in the trees, in 10 m or more most of the time (mean across Q4–Q6: 62%). By comparison, Eska was still roughly 40% of the time on the ground (mean across Q1 to Q6). However, Figure 9a–c and Appendix A indicate a steady decrease of time on the ground for all juveniles, including Eska, from Q4 onward. By Q6, all juveniles spend more time above 10 m than on the ground (Figure 9a–c).

Inference statistic shows that all three juveniles increase the time they spend in the higher stratum categories (5 m and more) at the expense of time spent on and near the ground (see Appendix A; Amalia: χ^2^ = 830.7, df = 6, *p* < 0.001; Eska: χ^2^ = 2249.6, df = 15, *p* < 0.001; Cantik: χ^2^ = 1364.7, df = 15, *p* < 0.001). Again, this is a trend towards the sort of behaviour we would expect in wild adult orangutans.

### 3.5. Feeding Competence

Orangutans cannot survive in freedom unless they know what to eat in the forest, where and when to find food and how to access hidden or protected foods. We analysed how much time our rehabilitant orphans spent eating and processing different types of food (Figure 10). While the orangutan orphans spent all daylight hours roaming the forest and continuously foraged during this time, we still supplemented food on platforms in the forest school, mornings and evenings. The provided diet leans towards vegetables (e.g., green beans, cabbage, carrots), but small quantities of fruit (e.g., melon, banana, pineapple) are also given. Soy products and boiled eggs are offered as protein sources. These ‘non-forest foods’ are relatively richer in water and calories than non-domesticated plants and are eaten within relatively short time. Such ”non-forest foods” make up only between 3–6% of scans per individual. The category ”from caretaker” refers to forest foods, which the caregivers harvested and offered to the rehabilitants to model food choices and preparation of embedded foods. Because infants needed this modelling more, eating ”food from caregivers" ranged from 3–6% for infants, and only from 0.04–3% for juveniles who were already more proficient. Infants spent also more time eating invertebrates (3–5%) than did juveniles (0.07–0.3%). Likewise, “left-overs” were eaten more often by infants (3–7%) than by juveniles (0.02–2%)—this category refers to food (mostly fruit or seeds) on the ground, either because it fell due to ripeness or because it was dropped by other animals including peers. Water, soil, unidentified or uncategorisable foods were rarely consumed.

All rehabilitant orangutans consumed mainly forest foods, which they found themselves (juveniles: mean of 92%, range 89–96%, infants: mean of 82%, range 81–83%). For plants, this category can be further broken down into more specific categories (Figure 10, right side). Again, independent of the rehabilitant′s age, all orphans ate fruit most often (mean of 73% for juveniles, range: 67–78%; mean of 73% for infants, range: 68–79%). The non-fruit/non-seed categories are important because they represent the so-called fallback foods on which an orangutan′s survival depends during lean seasons. For non-fruit organs of plants, juveniles and infants had different preferences. The juveniles ate young leaves (mean: 9%, range: 8–10%), pith (mean: 6%, range: 5–7%), seeds (mean: 6%, range: 5–8%) and bark (cambium; mean: 4%, range: 2–6%). The infants ate primarily young leaves (mean: 8%, range: 4–11%) and flowers (mean: 6%, range: 5–6%), but also pith (mean: 5%, range 3–6%), and there was more variability in their choices of plant parts than in those of the older ones. Young leaves and also the much less favoured mature leaves are easily accessible to all age groups and readily consumed (mature leaves: juveniles mean 0.04%, range: 0–0.1%, infants mean: 1.5%, range: 1–2%). While extractive foraging of pith, seeds and bark characterized the fallback food choices of older orphans, flowers and invertebrates (infants: mean = 4%, range: 3–5%; juveniles: mean = 0.2%, range: 0.07–0.3%) were utilized more by the smaller-bodied infants.

## 4. Discussion

To document individual orangutan′s development of survival skills, we recorded spontaneous behaviours of wild-caught orangutan orphans during their rehabilitation for later re-introduction into natural habitat. The rehabilitation environment was as close as possible to conditions in the wild with the orphans being immersed in a natural forest during the entire day, and only limited exposure to human caregivers and human culture. When preparing ape orphans for later release, free wild apes serve as standard, as the norm we wish rehabilitants to reach. Therefore, we collected data according to behaviour categories that are well established in the study of wild orangutans [20] and also for post-release [16,21].

### 4.1. Activity Budgets

Because rehabilitants were supplied with additional food regardless of seasonal fluctuations in forest productivity, the most appropriate comparison is with free-living orangutans who live in conditions with rather regular food supply. In their review paper Morrogh-Bernard et al. [20] (Figure 8.1) present data on five sites in Sumatra and Borneo with regular food supply. Activity budgets of adult wild orangutans usually represent only crude categories, feeding, resting, travelling and “other”. At all sites, orangutans exhibited the same ranking of these daily activities. In their 12-h daylight period, food consumption and manipulation take up by far the most time—more than 6 h. Next is resting with roughly 2–3 h per day, followed by travelling for which only 1–2 h are allocated. Less than half an hour, on average, is spent with “other” behaviours, including social interactions.

Amongst our rehabilitants only eight-year-old Amalia’s activity budget paralleled wild adults’ activity allocation with a predominance of feeding, followed by resting and travelling and only very small portions of social interactions and other behaviours. All orphans, regardless of age, spent most of their day with feeding and foraging. However, apart from Amalia, they tended to travel more and rest less than their adult wild counterparts. To discern whether this is an age effect, we compiled what activity-related data were available in the literature to enable a comparison with mother-reared wild immatures (Table 6). In each free-living immature the time spent resting exceeded the time spent travelling. These comparisons clearly indicate a reduction of resting time in our rehabilitant orphans compared to wild mother-reared age mates. Lack of resting, potentially restlessness, could be a matter of concern, indicating that peer-raised cross-fostered orphans do not get enough rest.

Unlike mother-reared infants, rehabilitant orphans have many social opportunities being immersed in a much more social environment than wild age mates. Potential social partners include several human allo-mothers and similar aged conspecifics. One would therefore expect orphans to spend much time on social interactions, especially play. This enforced sociality may also translate into increased locomotion, reflected in heightened travel times and reduced resting. In addition, the provisioning of extra food to compensate for still inadequate feeding skills guarantees a positive energy balance even with increased levels of activity and stimulation. Consequently, it remains yet unclear whether the reduced resting times pose a problem or not. If the orphans rested less because they received more social stimulation in their rehabilitation environment than wild immatures do, this could not only explain the rescued resting time, but might also lead to higher tolerance for excessive stimulation. Raised tolerance of stimulation would in turn lead to novelty seeking and risk-taking behaviours, which would tend to lead rehabilitants into danger [24,25,26]. An association between insecure attachments and risk-prone behaviour has been established in humans [27,28,29]. This is in line with physiological research that established a blunting of stress responses as a result of chronic stress in various mammals [30,31,32,33,34].

Play, both solitary and social, is a typical stimulation-seeking behaviour, and under stress and over-stimulation mammals cease to play. In wild-born orangutan infants, play can take up to 45% of their total activity, but the variation in play time among mother-reared infants appears immense and is not clearly associated with age (Table 6). While the rehabilitant infants in JP fell within the range of this variation, juveniles Eska and Cantik played multiple times more often than wild-born age-mates at Ketambe. Wild immatures played most often alone, followed by social play with peers, and played least with their mothers [35] (p. 196). By contrast, our rehabilitant orphans were engaged in social play with peers more often than in solitary play (similar or even higher rates), and they played least with their surrogate mothers. Thus, the rates and patterns (solitary vs. social play) of rehabilitant orphans in this study diverged from those of wild mother-reared immatures. This suggests that orphans were not over-stimulated to an extent where they no longer sought additional entertainment by exploration and play. However, the high rates for social play together with the reduced resting times could be an indication of elevated stimulation thresholds that might, in turn, induce human-raised orphans to adopt risk-seeking and neophilic behaviours. Several studies have found that wild orangutans are characterised by marked neophobia and risk aversion, while orangutans exposed to human culture are neophilic and apparently unaware of risks [36,37]. It seems plausible that this pattern is caused by habituation to markedly higher stimulation levels in a human environment than is typical for the environment of evolutionary history.

### 4.2. Development towards Independence

Orangutans are the mammal with the slowest life history and the longest maternal dependency [2,38]. In the intimate dyad of mother and infant security is provided. Psychological attachment bestows the infant with felt security that paves the way to future psychological resilience [25]. Skills and cognitive processes are developed through sharing and observational learning. It is obvious that maternal loss in this delicate period has the potential to fatally derail an immature’s development. Studies on humans and non-human great apes have shown that sensitive and responsive care will not only make mothers a safe base for the infant but will also enable exploration behaviour, and thus facilitate learning and cognitive performance [39,40]. In short, independence requires security with a safe base [41]. To understand these interrelationships, it is helpful to tease apart the facets of maternal investment and the functions the mother assumes in her infant’s development.

An important difference between rescued orangutan orphans and mother-reared wild orangutans is the closeness of mother and offspring. Up to the age of four years, mother-reared infants spend at least 80 percent of the time in less than 10 m distance from their mothers (Table 7). The younger the infant is the more time it spends within arm’s reach of the mother, including clinging to her body (Table 6). During the first year of life, the infant is rarely more than 2 m away from mother. Yet, in orangutans, the mother-infant bond is not only close and intimate, but also exclusive. Before weaning, wild immatures have no contact with other adult or subadult orangutans, except, occasionally, with an older maternal sibling [42]. They may touch and play with another immature if their mother meets another female who is also accompanied by her offspring. However, unrelated orangutans extremely rarely affiliate, or touch infants accompanied by their mothers [35,43]. Exclusiveness and spatial proximity to a primary caregiver are associated in mother-reared orangutans, together they spell intimacy. For orphans, however, they occur independently.

#### 4.2.1. Attachment as Closeness: Spatial Proximity to Surrogate Mothers

The data on spatial proximity in Table 7 show a large variability between infants, ranging e.g., for 2-year-olds from 16 to 95 percent of time spent within 2 m from mother. Compared to the mother-reared age mates at Ketambe and Danum Valley the rehabilitant orphans spent dramatically less time close to surrogate mothers (Table 7) but are within the range of infants at Suaq. We presented data on orphans aged between 19 months (Gerhana) at the beginning of the data collection and nine years (Amalia) at the end. The distances to human surrogate mothers reflect orphans′ ages. Only the eldest, Amalia and Eska, were almost never cuddling and within 1 m of a caregiver. However, Eska was still clearly attached to his caregivers, spending much time within 5 m of them. Cantik, on the other hand, while occasionally engaging in body contact was also often more than 10 m away from any caregiver, more often than Eska. Amalia, however, spent most of her time in over 10 m distance from caregivers, and appeared, at times, to deliberately elude them. Only Amalia got herself “lost” from sight repeatedly and was found after hours of searching in areas quite outside her usual range, thus suggesting an unparalleled desire for privacy or independence. In this Amalia fits the pattern of juvenile solitariness that appears to be typical of Bornean orangutans [35] (p. 198/9). Thus, while the orphans do show developmental trends that reflect what is known for mother-reared wild immatures they started out precociously with much lower portions of daytime spent in body contact and close proximity than their wild age mates.

#### 4.2.2. Attachment as Exclusiveness: Choice of Social Partners

After weaning, juveniles will start to be more distant, but stay in association with mothers, i.e., coordinate movements to remain in ca. 50 m distance, until reaching puberty at the age of 9–10 years onward [45,46]. During their gradual development towards full independence, from 6 until ca. 15 years of age, immature orangutans are relatively more social than mature ones, and a lot of their social time is spent with play. In this period, they also interact with unrelated juveniles, subadults and adult conspecifics [43]. This period is arguably the most social in their lives. Thus, the exclusiveness of the bond and the tight spatial association with the mother wane at the same time. Rehabilitant orphans, in contrast, are forced to divorce closeness from exclusiveness, because they must bond to several mothers who take turns in caring for them and they must share these mothers with other orphans. In the allo-maternal setting that prevails in rehabilitation programmes orangutan infants can still exercise some evolved preferences. They can choose between male and female caregivers and between human surrogate parents and same-aged conspecifics. We have shown above that being close to social partners waned in our rehabilitants in the same age period as it does in mother-reared immatures. What, then, about exclusiveness?

Overall, the time orphans spent on social interactions decreased with age, regardless of the type of interaction partner. The oldest infant, Kartini, had interaction frequencies already similar to the younger juveniles. All three females interacted less than their male peers. Whereas the infants interacted with human caregivers in 5–10 percent of daily activities, juveniles did so rarely. Among the humans acting as maternal surrogates, infants and 4-year old Cantik preferred female caregivers for close proximity and cuddling. Amalia, by contrast, generally eschewed female caregivers and preferred male caregivers on the rare occasions when she was in close proximity. The need for 24 h care and the 40 h week of a normal employment did not leave much room for exclusive surrogate mother to orphan bonds. Qualitative observations suggest that orphans were thwarted in their desire to form exclusive bonds: Some orphans showed a marked preference for a specific caregiver and exhibited increased self-confidence in that person’s presence. Other orphans, however, lacking such a preferred bond themselves, responded with clinginess and showed despondency when exposed to an orphan with his or her preferred caregiver. Competition for surrogate mothers was evident in contact conflicts and—a sometimes aggressive—displacement of rivals for body contact and attention from human caregivers. If caregivers leave, e.g., when shifts change, orphans will usually attempt to follow and need to be actively distracted. Indeed, some orphans′ behaviour is compatible with the indiscriminate attachment behaviour labelled in human infants Disinhibited Attachment Disorder of childhood [47] or Disinhibited Social Engagement Disorder [48,49] (for orangutans: [50]).

Thus, there is a decoupling of exclusiveness from closeness, which could indicate a loss of intimacy. This felt exclusiveness might be vital for wellbeing and a healthy psychological development and as a chronically unsatisfied need might find expression in, for example, a reduced ability to rest and relax, indiscriminate friendliness to any human in reach, or ambivalent reactions towards caregivers.

#### 4.2.3. Attachment Needs and Peers

Since the studies of Harlow and colleagues (for review e.g., [51]) it has been accepted that peer rearing improves mental health of primates growing up without mothers. Social competences are acquired during conspecific play and stimulation thresholds are established in a range similar to reference values established under natural rearing conditions [24,52,53,54,55]. While these insights are based on gregarious species, by inference, peer rearing has also become a ubiquitous component in rehabilitating orangutan orphans, at times intended to replace maternal care. However, in view of orangutans’ solitary life style and long inter-birth interval, growing up in daily association and interaction with age mates cannot be expected to come natural to orangutans. Indeed, anecdotes from peer-groups of intense competition and bullying abound. Nonetheless, close bonds between orphans may also develop and can lead to truly altruistic sharing [56].

To enable fissioning during the day without leaving orphans unattended, the ratio of caregivers to orphans is high in Jejak Pulang′s programme. As a result, of these managerial decisions, when meeting peers, orphans were always in the company of humans. When interacting with peers, infants did not seek out partners who were not already in their party, but rather chose between interacting and not interacting with the peer(s) in their party. Throughout the observation period, infants spent around 12 percent of their daily activities interacting with each other, more time than with humans (mean of 9 percent). We observed differences between infants regarding their ecological competences, with female Kartini showing the most advanced skills. To facilitate social learning, we therefore mixed daily parties in a rotational system. As a result, the formation of exclusive bonds between orphans was impeded. During the night, however, the same pairwise sleeping combinations were maintained throughout (Kartini with Gerhana, Gonda with Tegar). There were phases during which Gonda acted excessively assertive and bullied Tegar so that special measures needed to be taken to make the situation more bearable for Tegar, who also tended to suffer from ill health.

Different from the infants, older orphans (“juveniles”) took initiative and actively sought out or avoided each other. Within the group of the juveniles, it was Cantik who was usually the victim of competitive actions or too rough play. She managed such situations largely by avoiding proximity. A few times she was displaced from her night nest by Amalia or Eska. She responded by climbing down to return with her caregiver to sleep in the night cage, showing that at 5 to 6 years she still resorted to the use of “maternal” protection especially when tired. While we found that all three juveniles decreased their social interactions with peers over time, Eska retained some insecurity about sleeping in the forest alone. He was the one who most often accompanying caregivers to his night cage instead of sleeping in a tree nest. When he slept in the forest, he frequently slept not far from one of his peers and was regularly permitted to share Amalia’s nest.

In conclusion, the age-specific needs for close and exclusive bonding could not be matched. Infant orphans were precocious in that they spent less time in body contact and close proximity (generally below 10 percent) with surrogate mothers than did wild immatures with their natural mothers. Younger orphans exhibited a preference for female surrogate mothers. As rehabilitation managers, we had to decide on a trade-off between facilitating social learning from more proficient peers on the one side and, on the other side, enabling attachments between individual peers and protection of weaker orphans from bullying by others. That infant orphans associated within 5 m of each other relatively frequently suggests that peers do compensate to some extent for the relatively reduced proximity between orphans and human surrogate mothers. However, the type of interactions amongst peers differs from the type of interactions that mother-reared infants engage in. As a consequence, the nature of the social bonds between orphans, and thus their attachment histories, are markedly different from maternally reared wild orangutans. It is possible that this translates into a general lack of felt security, combined with a degree of boredom where human surrogate mothers simply fail to emulate the lifestyle of an orangutan mother.

### 4.3. Use of Forest Stratum

For a rehabilitant immature orangutan raised by human surrogate mother it is usually necessary to make a choice between normal orangutan behaviour, which implies being in the canopy, and staying close to one’s mother who provides security. Wild mother-reared immatures are not forced to choose and have regular access to maternal security and arboreal environment at the same time.

As expected, infants compromised by spending considerable amounts of time on the ground. Yet, as juveniles became more independent and spent less time with surrogate mothers they also spent less and less time on the ground. This development was mirrored in the time spent high up in the trees, which increased in parallel to the reduction of time on the ground.

It is noteworthy that all infants did make use of the highest forest stratum. While food attracts them to the canopy, it seems also that being high up and/or far from a safe base demands courage. In periods when Tegar was bullied by Gonda he would eschew being in trees while Gonda was using the canopy and would climb up after Gonda had returned to the caregivers. We attempted to familiarise the infants with the canopy by using professional tree climbing skills to hoist caregivers into over 10 m height where they could act as a safe base while infants explored, fed, peered at experts, competed or played [57].

### 4.4. Feeding Competence

As any other skill, feeding competence develops over time. Incisors are the first teeth the appear in orangutans, from ca. 5 months of age, just a little earlier than in humans. From this age on, orangutan infants start to ingest food. By the age of weaning, an orangutan juvenile’s diet overlaps by nearly 90 percent with that of his or her mother [58]. Deprived of mother’s modelling and tuition orphans have to rely on human surrogates with imperfect knowledge, different techniques for food extraction and lacking arboreal competences (but see [57]). The modelling of human surrogate mothers is complemented by trial-and-error learning and by social learning from other orphans (horizontal—oblique transmission). In JP’s rehabilitation programme, feeding competence can also be learned from more accomplished orphans Amalia (by Eska and Cantik) and Cantik (by Eska and the infants). Among the infants, Kartini was the first to show extractive foraging and therefore served as role model for the others.

All orphans ate fruit preferentially to other foods. It is noteworthy that during the fruit-rich season the older orangutans frequently failed to arrive at the platform for supper and breakfast. Instead, they made nests close to the trees on which they fed. In this way, they economised travel time and also showed that they satisfied their needs from forest foods alone. This pattern is typical also for competent wild orangutans.

To assess rehabilitation success the non-fruit/non-seed categories are more relevant because they represent the so-called fallback foods on which an orangutan’s survival depends during lean seasons. Caregivers were often modelling the consumption of fallback foods as these might save an orangutan’s life during periods of food scarcity, and previous programmes have shown that this is a task on which orangutan may fail after release [12,59,60]. Processing and opening protected foods, however, requires strength and dexterity that infants have yet to acquire. That maturation is important is supported by the observation that the oldest infant, female Kartini, was the one who first and most often endeavoured to splice open rattan or gnaw away bark to reach the cambium. Modelling represents an essential part of the education in forest school where the caregivers model processing food items. Modelling is combined with food sharing, as reflected in the “food from caretakers” results. Orphans are acutely curious when caregivers handle forest foods and ingest such foods without hesitation. Forest foods processed by caregivers are usually protected and need preparation or tools to become accessible. Splicing open thorny rattan shoots, or fruits of Castanopsis and palm nuts are typical for this category, but the items vary with the competence and physical strength of the orphan. Older orangutans were more proficient, therefore they received less food “from caretaker” than the infants who relied strongly on caregivers in eating such foods.

Interestingly, the variety of fallback foods utilised by infants was larger than in juveniles. Marzec [61] similarly found that newly released orangutans utilised a higher variety of foods than did orangutans released longer ago. She interpreted this as a honing of feeding skills towards increased efficacy. Lack of efficacy in finding foods would also explain the unusually high proportion of travel in the orphans′ activity budgets. Travel times would thus simply reflect increased search effort. Moreover, infants differed from juveniles in their choice of fallback foods. Importantly, the standard fallbacks of leaves (especially young ones), bark/cambium as well as figs and leguminosa were consumed by all orphans. Additionally, infants consumed flowers and invertebrates. The caregivers frequently offered termites and also modelled how to extract and eat them. By contrast, the juveniles ate more seeds. The distinction between seeds and fruit, however, was not always reliable. In these food choices we probably see not only the effect of improved knowledge, dexterity and increased physical strength of the older ones, but also the difference in economic harvesting for smaller and larger bodied orphans—patches of flowers or termite nests can satiate a small orphans of 10–12 kg (boys), but are unlikely to satisfy a juvenile of 27–32 kg (Amalia).

To summarise, while we found that rehabilitant orphans utilised fallback foods when fruit was scarce, we also observed indications that choice and processing capacities for fallback foods is subject to maturation effects, such as increased strength and dexterousness. Learning and physical changes during ontogeny seem to influence optimal foraging/feeding efficacy with the result that, regarding fallback foods, diets of infants and older orphans diverge. A predilection for fruit was universal and appears to be innate. While lack of knowledge seems to forestall optimal foraging effort, e.g., by increased travel times to compensate ignorance about where food might be found, rehabilitants appear to recognize once they strike a reasonable balance between time/effort invested and calorie intake relative to their own body weight.

## 5. Conclusions

### 5.1. Summary of Differences between Mother-Reared Immatures and Orphans in JP

Time budgets of rehabilitant orphans agreed with those of wild adult orangutans in the priority given to feeding. Only eight-year old Amalia′s activity budget reflected time allocation of adult Bornean orangutans for resting, travelling and social or “other” behaviours. Younger orphans rested less but travelled and interacted socially more. Between four and five years, this changed and orphans approaching juvenility increased time allocated for resting at the expense of time for social interactions.

Younger orphans had more variable diets than older ones. This could be the result of trial-and-error learning, similar to newly released orangutans, but also a reflection of different energy budgets for larger and smaller bodied immatures. In view of the still lacking feeding competence, it seems warranted to provision immature orphans with additional food, but older orphans indicated that they no longer needed supplementary food by not showing up at the feeling platform during fruit-rich periods.

Important deviations were found regarding social behaviours and the development of independence. These deviations are rooted in the different social settings. While wild mother-reared immatures rarely meet conspecifics and spend 100 percent of their time in close contact and proximity of their mothers, orphans showed different patterns. First, they spent less time in close proximity of a surrogate mother. Second, being brought up in an allo-maternal setting they were prevented from forming exclusive bonds with one primary caregiver. Third, they met and interacted with three to four conspecific immatures regularly and much more frequently than do wild immatures. However, embedded in a quasi-natural forest environment, older immatures started to reduce social contacts by spending more and more time far away not only from human caregivers, but also from conspecifics. At the same time, they increased their time in the canopy at the expense of time spent on the ground.

### 5.2. Changes over Time: Maturation and Rehabilitation

Our data set allows some tentative conclusions about age effects because we can cross-sectionally compare younger and older orphans, but we can also see the development over 18 months within each individual, longitudinally.

It is difficult to get good age estimates when infants are taken in. Teeth eruption patterns yield only rough approximations. Upon arrival, the physical condition of orphans is highly diverse, including starvation and malnutrition. In human care, orangutans quickly gain weight and tend to grow and mature considerably faster than in the wild. As a result, of these inaccuracies, age estimates are less reliable for younger immatures and overall our age estimates may be too young. We could not detect systematic patterns of changes implying developmental trends in this developmental interval. It was only in the comparison with the older orphans (“juveniles”) that patterns could be discerned. Overall, it appeared as if females would develop competences faster and pay better attention, whereas males were often scrounging—assertively taking over foods or using another’s nest.

Interestingly, in the nine months from Q4 to Q6 several changes in the juveniles′ behaviours occurred simultaneously. They rested more, interacted less with each other, spent more time farther away from caregivers and more time higher up in the trees. When the changes in Q3 set in, Cantik was aged approximately 5, Eska ca. 6 years. At this age, for mother-reared infants, weaning might commence. We thus see a shift, from infant-like dependence towards reduced dependence and increased solitariness in juvenility, associated with an increase in forest-focused activities. It is likely that these age-appropriate changes were triggered by Amalia acting as a role model, in combination with the immersion in the forest, which provided many opportunities for orangutan-appropriate behaviours to develop in the absence of distractions by human peculiarities and commodities.

In conclusion, the patterns observed in our juveniles with respect to their behaviour, space use and feeding competence indicate that our rationale of aligning the rehabilitation measures with the natural ontogeny, “orangutanising” the human caregivers and maintaining a non-contact policy is conducive to prepare orphaned orangutans for later re-introduction into natural habitat, independent of humans. Providing opportunities for social learning from more proficient social partners—human as well as conspecific—is important for skill acquisition as frequent peering followed by exploration and copying indicate. However, if embedded in a natural orangutan environment, characterised by good natural forest, role models and the presence of friendly conspecifics, orphans also spontaneously develop appropriate behaviours that will stand them in good service upon re-introduction to freedom in a tropical rain forest.

## Figures and Tables

**Figure 1 animals-11-00767-f001:**
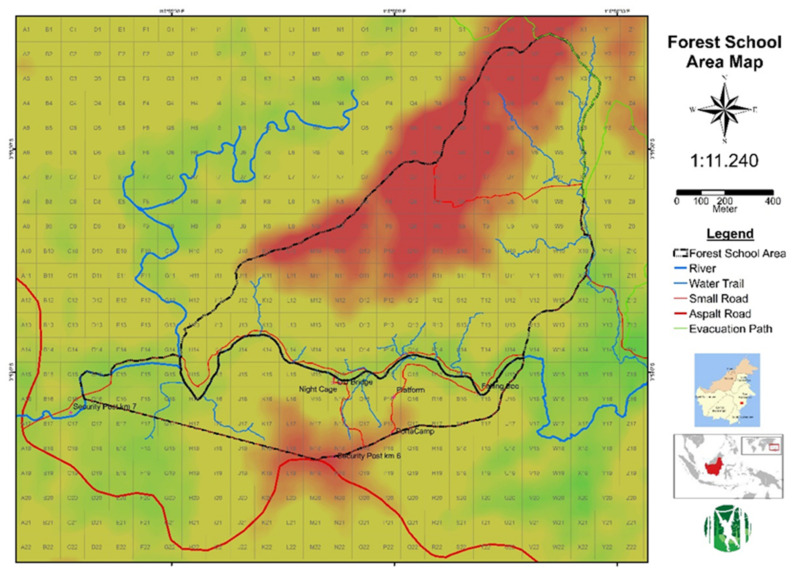
Topographic map of the Forest School (outlined in black) with locations of supporting structures Portacamp (for young orphans) and Post Sungai (for older orphans) south of river Sakakanan, and the quadrants enabling easy communication about current location of orangutans and people. Quadrants are 100 × 100 m in size.

**Figure 2 animals-11-00767-f002:**
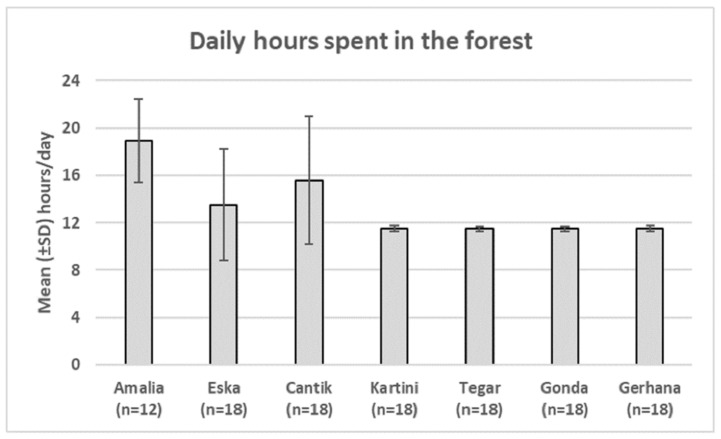
Mean time each rehabilitant spent in the forest per day. The three juveniles started to sleep in the forest increasingly often, therefore the mean can rise above 12 h of daylight. Means are reduced by the inclusion days when the orangutans had to be placed in cages, e.g., due to flooding. N refers to the number of months for which data are presented.

**Figure 3 animals-11-00767-f003:**
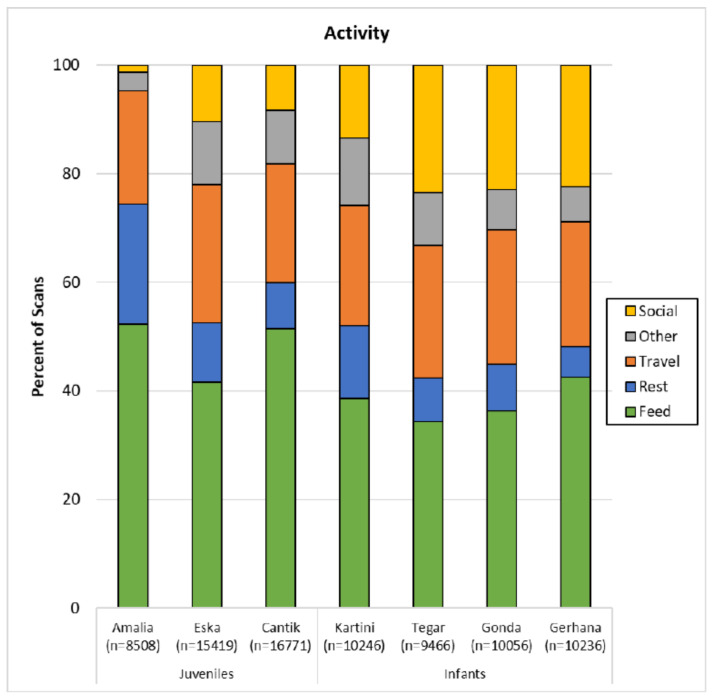
Activity budgets per subject over 18 months (Amalia: over 9 months). N refers to number of scans.

**Figure 4 animals-11-00767-f004:**
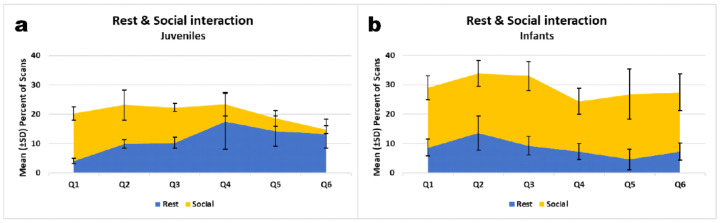
Mean proportion of activity budget spent on resting and social interaction for juveniles (**a**) and infants (**b**): Trends per rehabilitants over 6 quarters. Same data base as in Figure 3.

**Figure 5 animals-11-00767-f005:**
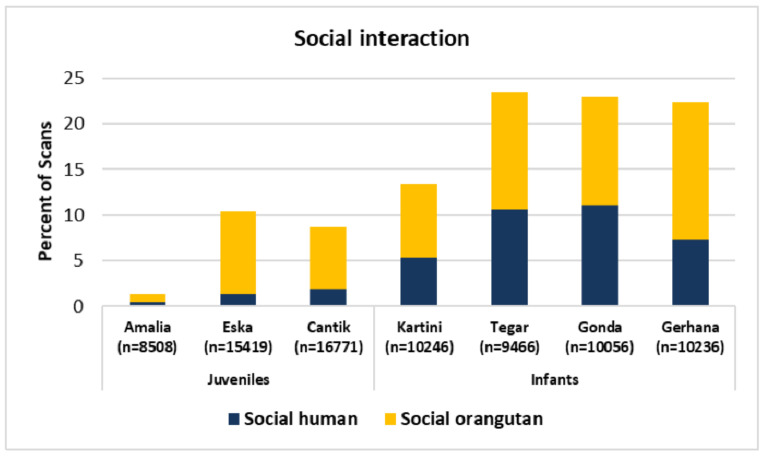
Proportion of the activity budget that was spent on social interactions with human caregivers vs. conspecific peers. Same data base as in Figure 3.

**Figure 6 animals-11-00767-f006:**
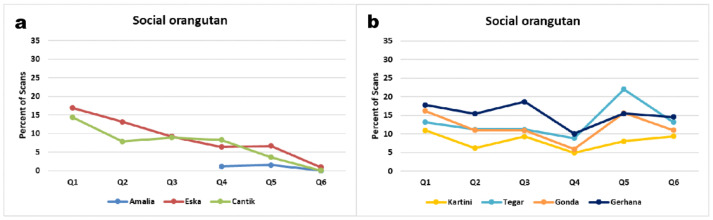
Development over time of social interaction frequency per juvenile (**a**) and infant (**b**) summarized across all conspecific peers. Same data base as Figure 3.

**Figure 7 animals-11-00767-f007:**
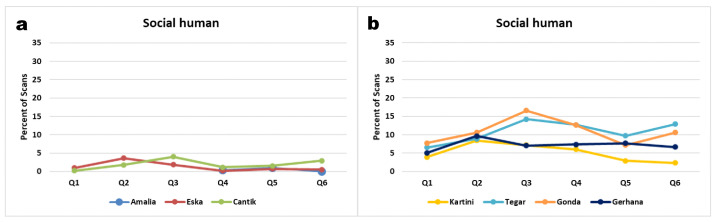
Development over time of social interaction frequency per juvenile (**a**) and infant (**b**), summarized across all human interaction partners. Amalia’s line (a, blue) is not visible, because it coincides with Eska’s (red, Q4–Q6). Same data base as in Figure 3.

**Figure 8 animals-11-00767-f008:**
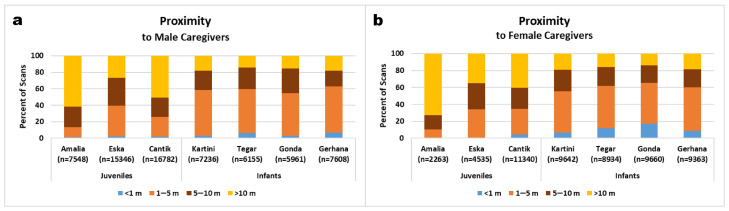
Proximity to male (**a**) and female (**b**) caregivers per subject over the entire observation period of 18 months (Amalia over 9 months).

**Figure 9 animals-11-00767-f009:**
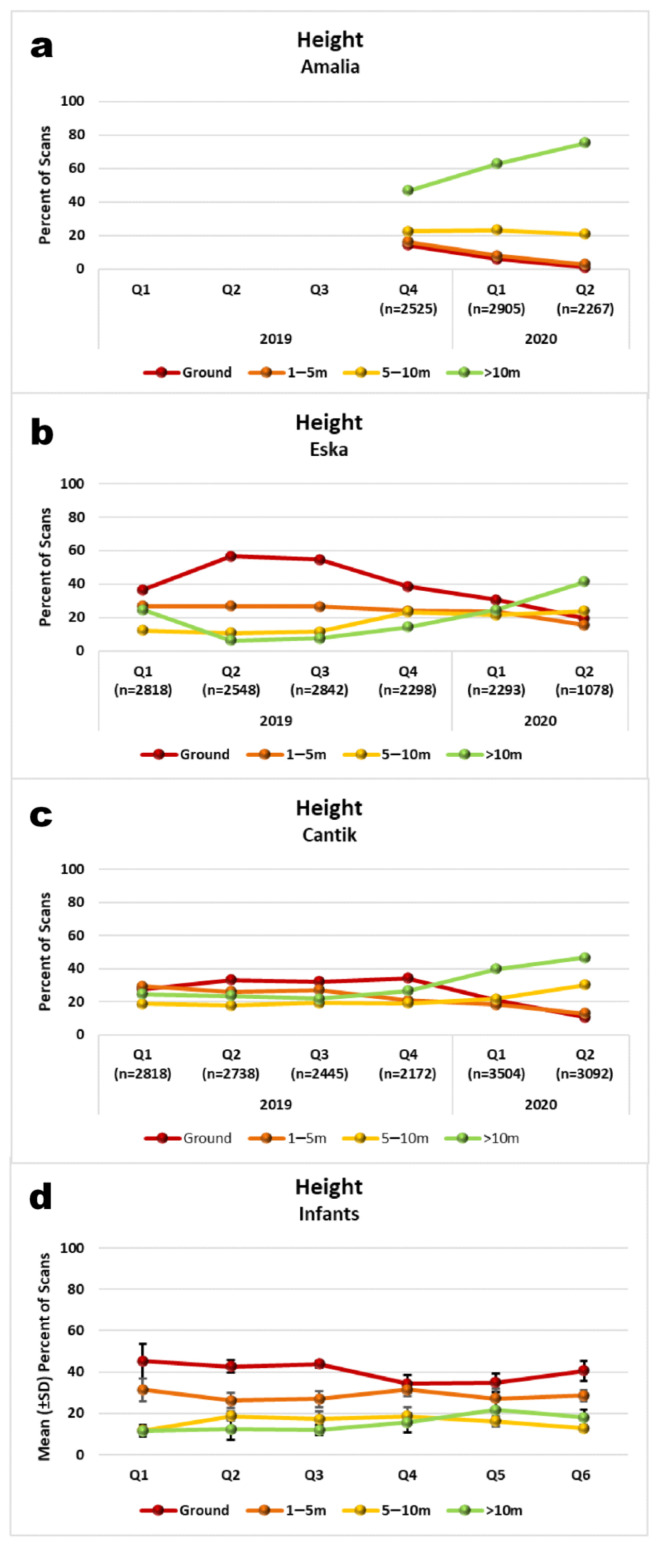
Use of forest stratum by individual juveniles (**a**–**c**) and means across all four infants (**d**).

**Figure 10 animals-11-00767-f010:**
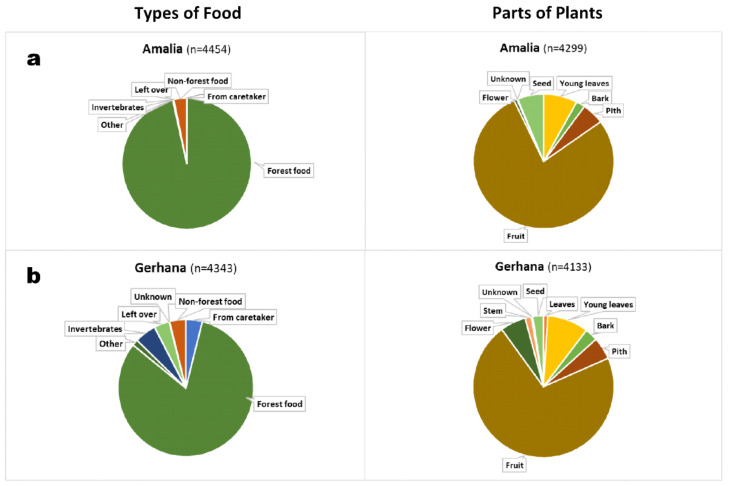
Types of food eaten (left) and plant organs consumed (right) by (**a**) the oldest rehabilitant Amalia (8–9 years), and (**b**) the youngest, Gerhana (19–37 months).

**Table 1 animals-11-00767-t001:** Biographic information on subjects in the forest school.

Name	Sex	Year of Intake	Est. Age at Intake	Est. Age at Start of Study Period	Level
Amalia	Female	2017	6 yrs	8 yrs	FS3
Eska	Male	2017	4 yrs	5 yrs	FS2/3
Cantik	Female	2017	3 yrs	4 yrs	FS2/3
Kartini	Female	2018	17+ mo	26 mo	FS1
Tegar	Male	2017	12 mo	27 mo	FS1
Gonda	Male	2017	8 mo	25 mo	FS1
Gerhana	Male	2018	9 mo	19 mo	FS1

Amalia started to attend forest school regularly from July 2019.

**Table 2 animals-11-00767-t002:** Total number of scans per orangutan per period and over all periods (sum).

Focal Orangutan	Q1 (Jan–Mar 2019)	Q2 (Apr–Jun 2019)	Q3 (Jul–Sep 2019)	Q4 (Oct–Dec 2019)	Q5 (Jan–Mar 2020)	Q6 (Apr–Jun 2020)	Sum
Amalia	X	X	X	2625	2912	2971	8508
Eska	2820	2545	2848	2295	2305	2606	15,419
Cantik	2803	2744	2462	2200	3455	3107	16,771
Kartini	1747	2251	1523	1679	1414	1632	10,246
Tegar	1664	2225	1391	1517	1173	1496	9466
Gonda	1805	2412	1598	1538	1188	1515	10,056
Gerhana	1727	2286	1725	1553	1249	1696	10,236

Note that the number of scans per quarter is deviating from the maximum possible number, as sometimes observations had to be cancelled e.g., due to heavy rainstorms, flooding or because Amalia disappeared and had to be searched for by caregivers and observers while Eska and Cantik had to stay in the cages.

**Table 3 animals-11-00767-t003:** Inter-individual comparison of the activity budgets.

Activity	Amalia	Eska	Cantik	Kartini	Tegar	Gonda	Gerhana
Feed	**13.1**	***−2.6***	**16.7**	***−6.8***	***−12.9***	***−10.2***	−0.8
Rest	**32.5**	1.0	***−8.4***	**8.6**	***−7.5***	***−6.3***	***−15.3***
Travel	***−4.6***	**5.5**	***−3.7***	***−2.3***	**2.1**	**3.1**	−0.6
Other	***−17.4***	**10.3**	**3.2**	**11.1**	**2.2**	***−5.4***	***−8.9***
Social	***−31.2***	***−11.8***	***−19.9***	−1.5	**24.6**	**23.8**	**22.6**

Values refer to standardized residuals (r). Values indicated in bold refer to significantly higher values than expected (i.e., standardized residuals of ≥2); values indicated in bold and italics refer to significantly lower values than expected (i.e., standardized residuals of ≤−2).

**Table 4 animals-11-00767-t004:** Percent of scans infants spent in proximity of one another (within 0–10 m).

Proximity	Kartini	Tegar	Gonda	Gerhana
Kartini	X	82.7	80.7	85.6
Tegar	84.3	X	85.3	85.1
Gonda	84.3	85.6	X	82.9
Gerhana	87.9	81.8	82.8	X

**Table 5 animals-11-00767-t005:** Percent of scans juveniles spent in association with each other (within 10–50+ m).

Association	Amalia	Eska	Cantik
Amalia	X	68.3	67.7
Eska	71.6	X	45.0
Cantik	76.4	53.7	X

**Table 6 animals-11-00767-t006:** Comparison of activities among mother-reared infants and our orphaned orangutans.

Activity/Clinging	Suaq ^(^^1)^		Jejak Pulang		Ketambe ^(2)^				
	1.5y *n* = 1 male	3y *n* = 1 female	1.5–3y *n* = 4	4–7y *n* = 2	1y *n* = 1 male	2y *n* = 1 male	3y *n* = 1 male	5y *n* = 2	6y *n* = 4
*Activity*									
feeding	7%	22%	34–42%	42–51%	33%	25%	49%	40–53%	50–68%
travelling	17%	13%	22–25%	22–25%	9%	2%	21%	18%	5–26%
resting	33%	17%	6–13%	8–11%	40%	55%	28%	29–39%	19–35%
playing	40%	45%	16–20%	14–18%	21%	21%	2%	2%	1%
*Clinging*	X	X	X	X	33%	45%	5%	1–8%	1–2%

^(1)^ [22]; ^(2)^ [23]. X: no data available. “Travelling” throughout is defined as independent locomotion and does not include being carried.

**Table 7 animals-11-00767-t007:** Comparison of spatial proximity between mother-reared infants and our orphaned orangutans.

Proximity	Suaq ^(1)^ *n* = 13 Immatures^(^*^)^			Jejak Pulang		Ketambe ^(^^2)^					Danum Valley ^(3)^ *n* = 6 Immatures ^(^*^)^		
	1–2y	2–2.5y	4y	1.5–3y *n* = 4	4–7y *n* = 2	1y *n* = 1	2y *n* = 1	3y *n* = 1	5y *n* = 2	6y *n* = 4	1y	3y	4y
0–2 m	25%	16%	X	7–17% ^a^	2–3% ^b^	83%	92%	31%	30%	36%	95%	50%	23%
2–10 m	75%	84%	80% ^e^	69–74% ^c^	48–71% ^d^	17%	8%	59%	65%	52%	5%	40%	55%
10–30 m	Never	Never	20%	14–19%	27–51%	Never	Never	9%	5%	10%	Never	10%	22%

^(^^1)^ [22]; ^(2)^ [23]; ^(3)^ [44]. ^(^*^)^ Variable numbers of individuals per age and category.^a^ <1 m to female caretakers; ^b^ <1 m to male caretakers; ^c^ 1–10 m to female caretakers; ^d^ 1–10 m to male caretakers; ^e^ 0–10 m distance. X: no data available.

## Data Availability

Data are reported in the Results and Appendix A sections of this paper. Upon request, additional information can be obtained from the corresponding authors.

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
