# Peer review of "Learning to Be an Orangutan—Implications of Life History for Orangutan Rehabilitation"

_animals, 2021, doi:10.3390/ani11030767_

Round 1
Reviewer 1 Report
This article addresses a critical issue relating to the ability to successfully rehabilitate orphan orangutans. I would like to commend the authors for dedicating themselves to such an important mission. It is very clear that a lot of thought is integrated into every aspect of the animals’ care.
In general, I find the article interesting and well written. The problem is clearly stated at the beginning of the introduction, and I have to say that I loved the direct and sometimes harsh critique the authors had on current practices.
One aspect that I found to be missing is more background information pertaining to the animals. I realize that this is often not available, but I encourage the authors to include it if they do have access to this. It will be interesting to interpret some of the inter-individual differences in behavior in light of this information. If this kind of information is lacking, I suggest the authors will add a comment relating to the potential influence that various experiences may have on the observed behavior.
Moreover, I think that the authors should emphasize the small sample size and interpret the results with a high degree of caution.
There are some grammatical issues, in particular, with regard to the use of correct tenses that require some revisions.
Also, some of the terminology used can be seen as anthropogenic. I highly recommend the authors to use descriptive information regarding the behavior of the animals and suggest an interpretation, emphasizing that it is an interpretation (e.g., despondency, jealousy, friend, courage).
My detailed comments are below.
17 – please change recued to rescued.
88 – please use either orangutan or OU throughout.
94-96 – I would not use the word shortcomings twice.
103 – please change to ‘enable’.
126 – please define random changes.
156 – please remove the hyphen.
Figure 2: Please change the y axis to a regular number format. It can be confusing as it looks like the time rather than the number of hours.
193 – please specify the method and statistic test used to evaluate IORs.
201 – please specify what were the ‘routine procedures’.
204 – please specify examples.
205 – please define ‘ecological competences’.
216 – please change to orangutan. I like the title used in the parenthesis more than the one that precedes them. To clarify, infants were observed 2 full days/month (i.e., 6 per Q), each full day represent 10 hrs. observation, and each hour observation had 30 scans (2 min intervals) – this results in 6d X 10 hrs. X 30 scans = 1800 scans (9 X 12 X 30 = 3240 scans for juvies). Please explain the deviations from these numbers that appear in Table 2.
241-242 – I find this suggestion a bit too strong with regard to the very small sample size.
250-251 – please rephrase. It reads somewhat strange.
309-316 – this paragraph belongs in the Discussion.
341-342 – again, the interpretation would fit better in the discussion. It would also be beneficial to add more info regarding the reason for which caregivers kept their distance from Eska.
344 – please be consistent. There are instances where you write ‘Table’ and others where you use ‘Tab’.
355-357 – this part is cumbersome. Please rephrase. As mentioned before, I find these lines better suited for the Discussion.
392-393 – please rephrase without using ‘it’ (where and when to find it).
Figure 10 – this figure is hard to read. I suggest removing some of the variables or lumping them together.
417-418 – since the percentages brought here refer to the proportion of types of food, this should be rephrased to something along the lines of: All rehabilitant orangutans consumed mainly forest foods, which they found themselves (juveniles: mean of 92%, range 89-96%, infants: mean of 82%, range 81-83%).
Table 5 – the title suggests that the Table includes data for orphans. It will be useful to have their data as well and examine the magnitude of differences.
486-487 – unclear. It seems like you want to say that since they rest less, they are more likely to encounter danger, but I am not sure.
493-512 – one of the major differences between orphans and mother-reared animals is the availability of partners to interact with. I would think that this is another factor that affect the differences in play. Also, I am curious if the authors think that it might be better to limit the social interactions available for the orphans in order to avoid neophilia.
516-520 – please provide references.
526 – please change ‘she’ to ‘the mother’ or ‘the dam’.
527 – child?
537 – I suggest using a different word than ‘friend’.
549-550, 552 – should be Table 6, not 5.
558 – ‘while perfectly able to enjoy a cuddle’ this is cute, but please rephrase to a manner more suitable to a scientific journal.
562 – I recommend changing ‘evincing’ to ‘suggesting’.
600-601 – please rephrase.
620-623 - please rephrase.
4.2.3. – I suggest adding interpretations and suggestions regarding the impact of abnormal rearing on the ability of orphans to thrive following their release.
671 – please change the word ‘mom’.
691 – please change ‘babies’ to ‘infants’ or ‘neonates’.
692- his or hers.
694 – deplorable seem a bit harsh.
769-772 – would the authors consider suggesting supplementing food for the infants only if there is a clinical issue relating to weight gain? This might increase motivation to search for food and gain more experience.
806-814 – I find this section unnecessary as it is a mere repetition on the results. I suggest removing.
820 – please change orang hutan
841 – please remove ‘or temptations?’.
Table S2-S4 - is there a different way to denote significant higher or lower values than the color scheme?
Reviewer 2 Report
Dear Authors,
First, thank you for submitting this interesting article. The work is generally well presented, with relevant use of graphs to demonstrate key points and good support in the form of citations throughout the work. I found the work to be interesting and useful contribution to the field of great ape literature. The work is well written with no obvious grammatical errors.
I have added a few minor comments on a PDF version of the paper which you should be able to download here. The two key areas that could be improved are as follows:
Life history. As life history is covered in the title of the paper, it is important to introduce the fundamental concept: for example, r / K selected traits. This only needs to be a brief introduction to the concept, but will make it much clearer to readers what is being measured: in this case, the extended period of parental care.
Length. The paper is extensive at 24 pages plus the appendix, and this may present some challenges to readers. While Animals does not have a strict word limit, I would suggest substantially reducing at least the conclusion to summarise only the key points from the paper.

Author Response
Reviewer 2
Dear Authors,
First, thank you for submitting this interesting article. The work is generally well presented, with relevant use of graphs to demonstrate key points and good support in the form of citations throughout the work. I found the work to be interesting and useful contribution to the field of great ape literature. The work is well written with no obvious grammatical errors.
I have added a few minor comments on a PDF version of the paper which you should be able to download here. The two key areas that could be improved are as follows:
Life history. As life history is covered in the title of the paper, it is important to introduce the fundamental concept: for example, r / K selected traits. This only needs to be a brief introduction to the concept, but will make it much clearer to readers what is being measured: in this case, the extended period of parental care.
- We included a respective paragraph in the introduction
Length. The paper is extensive at 24 pages plus the appendix, and this may present some challenges to readers. While Animals does not have a strict word limit, I would suggest substantially reducing at least the conclusion to summarise only the key points from the paper.
- We substantially reduced the conclusion.
We would like to thank Reviewer 2 for his/her valuable comments and suggestions!

Reviewer 3 Report
In this study, Preuschoft and colleagues present ontogenetic, behavioral data across a group of seven orphan orangutans over a period of 1.5 years. Interactions and activities in the studied individuals differed significantly from wild-reared conspecifics, spending less time interacting with surrogates than wild-reared juveniles would spend with their biological mothers. Within older orphans, activity schedules more closely aligned with wild adults.
Overall, this is a well-written study that puts forward valuable data on the activity budgets of orphan orangutans within a surrogate setting. Within these limitations, the data are present cleanly and without excessive interpretation. While the application of these data, given the somewhat constrained study design, is not overwhelming, this study is nonetheless a valuable contribution to the field. My detailed comments can be found below.
Specific Comments
In general, the aims of the study are not clearly articulated beyond a brief description of the project at large. A more explicit statement on the application of these observations to future conservation efforts is necessary but, more pressingly, clearer a priori predictions of behaviors (and behavioral changes) within orphan orangutans would make this study substantively stronger.
To do this, it is necessary to amend the introduction slightly to present more data on typical behaviors among wild-reared orangutans, from which predictions of the impact of surrogacy upon activity budgets may be derived. As currently constituted, the three assumed possible changes listed towards the end of the introduction are somewhat loose and insufficiently specific to frame the findings presented later, particularly as the discussion and conclusions go on to contrast time budgets of wild orangutans against rehabilitated orphans. A brief summary of this is presented in the discussion, but this would be stronger if both increased in depth and moved earlier within the manuscript.
Though the data are presented in a logical and intelligible manner, I would recommend moving Table S2 into the main body of the manuscript, under results. Differences in activity budgets between individuals are frequently discussed and I found myself moving back and forth between the manuscript and supplementary materials many times. Tables S3 and S4 seem appropriate to remain supplementary.
The discussion, particularly section 4.2 (but prior to 4.2.1), introduces a lot of novel background data with regards to the species that require the appropriate citations. Moreover, some of this information is probably more appropriate within the introduction, as it is not so much discussing the data presented within this paper but outlining the life history of orangutans – which would seem more appropriate at the start, and not at the end, of a paper about orangutan development.
